

# Predicting stimulation-dependent enhancer-promoter interactions from ChIP-Seq time course data

Tomasz Dzida[1], Mudassar Iqbal[1], Iryna Charapitsa[2], George Reid[2], Henk Stunnenberg[3], Filomena Matarese[3], Korbinian Grote[4], Antti Honkela[5,6,7] and Magnus Rattray[1]

[1] Faculty of Biology, Medicine and Health, University of Manchester, Manchester, United Kingdom
[2] Chemical Biology Core Facility, European Molecular Biology Laboratory, Heidelberg, Germany
[3] Department of Molecular Life Sciences, Radboud University Nijmegen, Nijmegen, Netherlands
[4] Genomatix GmbH, Munich, Germany
[5] Helsinki Institute for InformationTechnology (HIIT), Department of Computer Science, University of Helsinki, Helsinki, Finland
[6] Department of Mathematics and Statistics, University of Helsinki, Helsinki, Finland
[7] Department of Public Health, University of Helsinki, Helsinki, Finland

Corresponding authors
Tomasz Dzida,
tomasz.dzida@gmail.com
Magnus Rattray,
magnus.rattray@manchester.ac.uk

## ABSTRACT

We have developed a machine learning approach to predict stimulation-dependent enhancer-promoter interactions using evidence from changes in genomic protein occupancy over time. The occupancy of estrogen receptor alpha (ER$\alpha$), RNA polymerase (Pol II) and histone marks H2AZ and H3K4me3 were measured over time using ChIP-Seq experiments in MCF7 cells stimulated with estrogen. A Bayesian classifier was developed which uses the correlation of temporal binding patterns at enhancers and promoters and genomic proximity as features to predict interactions. This method was trained using experimentally determined interactions from the same system and was shown to achieve much higher precision than predictions based on the genomic proximity of nearest ER$\alpha$ binding. We use the method to identify a genome-wide confident set of ER$\alpha$ target genes and their regulatory enhancers genome-wide. Validation with publicly available GRO-Seq data demonstrates that our predicted targets are much more likely to show early nascent transcription than predictions based on genomic ER$\alpha$ binding proximity alone.

## INTRODUCTION

Gene expression is dependent upon the binding of transcription factor (TF) proteins to genomic regions which regulate transcriptional initiation (*Nagarajan et al., 2014*). In eukaryotic cells, these regulatory genomic regions are referred to as promoters and enhancers. The transcriptional competence of DNA in eukaryotes is determined by its organization in chromatin. Chromatin structure is dynamically regulated at multiple levels, including ATP-dependent chromatin remodelling and histone modifications (*Bernstein et al., 2005*; *Bannister & Kouzarides, 2011*; *Zhu et al., 2013*; *Stasevich et al., 2014*). Enhancers

can act upstream or downstream of their target gene promoters and are often distal, separated by large inter-genic regions (*Schoenfelder et al., 2010*; *Sanyal et al., 2012*; *Shen et al., 2012*). Enhancer-promoter interactions require protein-mediated physical contact through formation of chromatin loops (*Tolhuis et al., 2002*). Although most contacts are intra-chromosomal, there are some interactions between loci from different chromosomes (*Fullwood et al., 2009*; *Li et al., 2010*; *Li et al., 2012*). Interactions can also exist as part of large multi-gene and multi-enhancer complexes (*Fullwood et al., 2009*; *Li et al., 2012*).

Recent progress in experimental techniques such as ChIA-PET, 3C and its derivatives 4C, 5C, and Hi-C (*Fullwood et al., 2009*; *Dekker et al., 2002*; *Hagège et al., 2007*; *Zhao et al., 2006*; *Dostie et al., 2006*; *Simonis, Kooren & de Laat, 2007*; *Van Steensel & Dekker, 2010*; *Nagano et al., 2013*; *Jin et al., 2013*) have mapped large numbers of chromatin interactions, including enhancer-promoter interactions. However, these methods are technically challenging and genome-wide methods, such as HiC, typically lack the resolution required to identify individual interacting enhancer elements. Some methods are also thought to produce a high false negative rate (in case of ChIA-PET, 5C; *Li et al., 2012*; *He et al., 2014*) or cannot be applied on a genome-wide scale (3C, 4C; *Simonis, Kooren & de Laat, 2007*). Capture-HiC methods have recently been developed (*Mifsud et al., 2015*; *Javierre et al., 2016*) to improve genomic resolution through focussing on predetermined genomic regions, e.g., promoters, and show promise but are not yet widely used. Data from these technologies can also be noisy and subject to various sources of bias which can be problematic to correct (*Van Steensel & Dekker, 2010*). In addition, the physical contact between two chromatin regions does not determine a functional interaction (*Shlyueva, Stampfel & Stark, 2014*) with stimulus-dependant behaviour of chromatin looping adding a further layer of complexity (*Drissen et al., 2004*; *Vakoc et al., 2005*). For these reasons, complementary approaches to infer enhancer-promoter interactions by exploiting readily available sources of genomic data, such as ChIP-Seq and RNA-Seq data, are of interest.

ChIP-seq experiments enable the discovery of the genomic location of transcriptionally relevant proteins such as TFs, RNA polymerase and modified histones. Multiple ChIP-Seq datasets can be combined with data from other relevant genomics assays to identify active promoters and enhancers using genomic segmentation algorithms (*Zhu et al., 2013*; *Ernst et al., 2011*). Others have also used ChIP-seq and RNA-seq datasets to infer enhancer-promoter interactions. For example, *Ernst et al. (2011)* used histone mark data from multiple cell-types to identify active enhancers and promoters from which enhancer-associated data was correlated with expression data from genes within 125 kbp to identify likely interactions. *Thurman et al. (2012)* used DNase I hypersensitivity (DHS) data from multiple cell-types to correlate and link distal DNase hypersensitivity sites (within 500 kbp) to those within putative gene targets. Similarly, *Andersson et al. (2014)* predicted enhancer-promoter links by correlating CAGE enhancer RNA to CAGE promoter RNA.

Approaches for discovering cell-type specific interactions include PreSTIGE (*Corradin et al., 2014*), RIPPLE (*Roy et al., 2015*), and the method developed by *Marstrand & Storey (2014)*. PreSTIGE uses a method based on the Shannon entropy to identify cell-type specific interactions between enhancers and genes using H3K4me1 and RNA-seq data respectively. The regions are linked within promoter-centric domains, bounded on each

side by the minimal distance of 100 kbp up to the first CTCF binding site from a TSS. RIPPLE uses ENCODE data from four cell-lines each with 11 ChIP-Seq datasets (RNA-seq, CTCF, RAD21, DNAse1, TBP and histone marks) to train a random forest classifier which predicts enhancer-gene interactions within 1MB distance. The features used are two joint binary vectors of presence/absence of dataset signal peak over a promoter and enhancer, correlation of entries of the vectors, as well as gene expression of the promoter controlled gene. *Marstrand & Storey (2014)* developed a method to aggregate RNA-seq data over genes and DHS data over $\pm$ 200 kb regions surrounding them for twenty different cell lines. The method searches through each gene and cell-line for unexpected DHS/RNA-seq ratios and once found, scans across the gene vicinities in search of causal, local DHS variabilities. Lastly, a method proposed by *He et al. (2014)* uses a random forest classifier to find enhancer-gene interactions. The method uses three features: evolutionary conservation, correlation of enhancer scores derived from histone marks from RNA-seq data, and an average of correlations between TF ChIP-Seq and gene expression across 12 cell-types. A distance constraint is also imposed to aid inference.

The majority of the above methods require data from multiple cell-types and therefore do not allow discovery of interactions given data from one cell-type. Most existing methods also assume a stringent distance constraint and are therefore unable to discover distal links beyond this constraint. Finally, these methods do not take into account evidence from time course data.

We show how ChIP-Seq time course data that reports TF and RNA polymerase occupancy at multiple time points after cellular stimulation can be used to predict enhancer-promoter interactions within chromosomes. We have developed a Bayesian classifier that combines evidence from the correlation of ChIP-Seq time course data at enhancers and across gene bodies with the genomic separation of interacting elements as features. We apply our method to time course data from MCF7 breast cancer cells after stimulation with estradiol and we benchmark performance against publically available ChIA-PET data from this system. We show that our method performs much better than association by proximity, identifying many more interactions than predictions based on proximity alone. Estrogen Receptor (ER-$\alpha$) and RNA polymerase (Pol II) ChIP-Seq time course data are shown to be highly informative for predicting interactions. We also stratify our predicted interactions to those that lie within Topologically Associating Domains (TADS; *Dixon et al., 2012*) and those that span TADs, showing that our classifier can make useful predictions in both categories. Finally, we use our predictions to provide a highly confident list of directly ER-regulated target genes in this system and validate it against a GRO-seq dataset. Our predicted targets are much more likely to show early nascent transcription than predictions based on genomic ER-$\alpha$ binding proximity alone and predicted targets are involved in many biological processes associated with breast cancer. Our model thus offers biologically meaningful insight into the early transcriptional response to ER-$\alpha$.

## MATERIALS AND METHODS

### Data preparation

The aim of our experiment was to uncover the early response to estradiol (E2) in MCF7 breast cancer cells. Our previous studies included only the Pol-II and RNA-Seq time course data from these experiments (*Wa Maina et al., 2014*; *Honkela et al., 2015*) and here we include additional ChIP-Seq datasets. The first step was to create a reference sample in a ligand free environment. For that, the cells were placed into estradiol free media for three days, which reduced the binding between ER-$\alpha$ and E2. The cells were then ready to be re-exposed to E2. Following the introduction of E2, the resultant changes were tracked by multiple ChIP-seq experiments. The experiments were performed at 0, 5, 10, 20, 40, 80, 160, 320, 640 and 1,280 min after the stimulation. Each ChIP-seq experiment was carried out with a different antibody to measure genome-wide changes in genomic occupancy of their specific protein targets. Specifically, the studied protein factors and histone modifications were: ER-$\alpha$, H3K4me3, and H2AZ (data available from GEO: accession GSM2467201). Other previously published data from the same set of experiments are available for Pol-II ChIP-Seq and RNA-Seq (GEO accession GSE62789 and GSE44800; *Wa Maina et al., 2014*; *Honkela et al., 2015*).

### Preparation of MCF-7 cells

The MCF-7 human breast cancer cell line originates from a 69-year old Caucasian woman and is estrogen receptor (ER) positive, progesterone positive (PR) and HER2 negative. Here MCF-7 cells (a clonal isolate obtained from the ATCC (catalogue number HTB-22) kindly provided by Prof. Edison Liu, Jackson Laboratories, Maine, USA) were grown in 15 cm plates to 80% confluency. Plates were then washed two times with PBS and overlaid with 20 ml of phenol-red free high glucose DMEM (Gibco) containing 2% charcoal stripped FCS (Sigma). After 24 h of incubation, the cells were again washed with PBS and fresh media containing 2% charcoal stripped FCS was added. This process was repeated over a three day period to generate cells devoid of estrogen. The time course (5, 10, 20, 40, 80, 160, 320, 640 and 1280 min) was initiated by replacing media with prewarmed media containing 10 nM E2. In addition, an untreated sample was included in the experiment as a zero time point.

### ChIP-seq protocols and methods

Cells were fixed for 10 min at room temperature by the addition of formaldehyde to a final concentration of 1%, after which glycine was added to a concentration of 100 mM. Cells were then washed twice with PBS and collected into 2 ml of lysis buffer (150 mM NaCl, 20 mM Tris pH 8.0, 2 mM EDTA, 1% triton X-100, protease inhibitor (complete EDTA free, 04 693 132 001; Roche, Basel, Switzerland), 100 mM PMSF). The lysate was sonicated for 3 $\times$ 30 s using a Branson ultrasonicator equipped with a microtip on a power setting of 3 and a duty cycle of 90%. Samples were cooled on ice between rounds of sonication. Alternatively, a Bioruptor sonicator was used (power high, 15 mins total, 30 s on 30 s off; total volume of sample—1 ml) to fragment chromatin. In either case, the resulting sonicate was centrifuged at 4,000$\times$ g for 5 min, an aliquot of 10% retained for input and the remaining material transferred to a fresh tube. Four mg of anti-ERantibody

(HC-20, rabbit polyclonal, sc-543; Santa Cruz Biotechnology, Santa Cruz, CA, USA), 2 µg of anti-RNA Polymerase II antibody (AC-055-100, monoclonal, Diagenode, 001), 3 µg of anti-H3K4me3 antibody (pAb-MEHAHS-024, rabbit polyclonal, Diagenode, HC-0010) and 2 µg anti-Histone H2A.Z (acetyl K4 + K7 + K11) antibody (ab18262, sheep polyclonal, Abcam, 659355) were added to the samples, which were then incubated overnight at 4 °C with rotation. Chromatin antibody complexes were isolated, either by addition of 10 µL of protein G labeled magnetic beads (Millipore Pureproteome protein G magnetic beads, LSKMAGG10) prewashed in lysis buffer or with 20 µL protein A/G beads (Santa Cruz Biotechnology). Afterwards, the complexes obtained with protein G magnetic beads were washed three times with lysis buffer, then reverse crosslinked in 0.5 ml 5 M guanidine hydrochloride, 20 mM Hepes, 30% isopropanol, 10 mM EDTA for a minimum of 4 h at 65 °C. Recovered DNA was then purified using a Qiaquick spin column and eluted in 50 µL of 10 mM Tris pH 8.0. Where protein A/G beads were used, the complexes were washed sequentially with three different buffers at 4 °C: two times with solution of composition 0.1% SDS, 0.1% DOC, 1% Triton, 150 mM NaCl, 1 mM EDTA, 0.5 mM EGTA, 20 mM HEPES pH 7.6, once with the solution as before but with 500 mM NaCl, once with solution of composition 0.25 M LiCl, 0.5% DOC, 0.5% NP-40, 1 mM EDTA, 0.5 mM EGTA, 20 mM HEPES pH 7.6 and two times with 1 mM EDTA, 0.5 mM EGTA, 20 mM HEPES pH 7.6. A control library was generated by sequencing input DNA (non-ChIP genomic DNA). Immunopurified chromatin was eluted with 200 µL of elution buffer (1% SDS, 0.1 M NaHCO3), incubated at 65 °C for 4 h in the presence of 200 mM NaCl, isolated using a Qiaquick spin column and eluted in 50 µL of 10 mM Tris pH 8.0. Libraries were prepared for Illumina sequencing according to the manufacturer's protocols (Illumina). Briefly, DNA fragments were subject to sequential end repair and adaptor ligation. DNA fragments were subsequently size selected (approx. 300 base pair (bp)). The adaptor-modified DNA fragments were amplified by limited PCR (14 cycles). Quality control and concentration measurements were made by analysis of the PCR products by electrophoresis (Experion, BioRad) and by fluorometric dye binding using a Qubit fluorometer with the Quant-iT dsDNA HS Assay Kit (Q32851; Invitrogen, Carlsbad, CA, USA) respectively. Cluster generation and sequencing-bysynthesis (36 bp) was performed using the Illumina Genome Analyzer IIx (GAIIx) according to standard protocols of the manufacturer (Illumina).

## Alignment to a reference human genome

Raw reads from the experiments were mapped onto the human reference genome (NCBI_build37) using the Genomatix Mining Station (version 3.5.2; https://www.genomatix.de/solutions/genomatix-mining-station.html) to enable further analysis. The sequencing depth, i.e., the total number of sequenced reads, was very similar for each dataset, however, on average only 81%, 76%, 67%, 61%, 64% of ER-$\alpha$, Pol-II (rep 1), Pol-II (rep 2), H3K4me3, and H2AZ ChIP-seq reads were mapped uniquely to the genome. The non-uniquely mapped reads were discarded from further analysis. Using the statistical criterion provided by MACS, we established that our sequencing depth allows

for no duplicates of reads, thus we discarded any duplicated reads as they are most likely an artefact in ChIP-Seq.

### ER-α binding locations

The MACS package (v2.0, *p*-value: 1e−7, no control, estimation of $\lambda_{\text{local}}$ off) *Zhang et al. (2008)* was used for peak-calling and applied to each of the $0, 5, 10, 20, \ldots, 320$ min time course datasets to estimate ER-α binding locations. The last two time points (640 and 1,280 mins) were not included as the number of ER-α mapped reads was found to be very low at these times compared to earlier times. Persistent co-occurring ER-α binding locations (i.e occurring at least twice across two time points after $t = 0$) were merged by a union operation (similar to the mergeBED method from BEDTools (*Quinlan & Hall, 2010*)), otherwise they were discarded. The method is illustrated in Fig. S1. Since our analysis is aimed at intergenic ER-α-bound enhancers, we ignored the consensus peaks which overlapped with either gene bodies or upstream 300 bp-long regions by which the genes were extended to account for a promoter region. This is a limitation of the data used and the method could potentially work with different ChIP-Seq data. With other enhancer-associated ChIP-Seq data then we could also potentially apply the method to intronic enhancers.

### Time-series construction

We calculated the mapped read counts for each individual time point ChIP-seq dataset over the consensus ER-α binding sites to create time series over enhancer regions for each of our antibodies. To normalise the counts, we divided each read count over the total number of uniquely mapped and non-duplicated reads across all time points and multiplied the resultant values by the total number of mapped reads in the $t = 0$ min dataset. We concatenated the normalised counts to produce time series for each ChIP-seq dataset. We refer to each enhancer time series as $X_{j,n}$, where $j \in J$ (number of intergenic enhancers) and $n \in N$ (number of time course ChIP-seq datasets). We repeated the process for the gene regions to create the analogous time series over gene regions, extending the genes by 300 bp upstream from their canonical TSS. We refer to each time series over gene as $Y_{k,n}$ where $k \in K$ (number of genes). We filtered out genes and intergenic enhancers from consideration if the total number of mapped reads across all time series was less than 30.

### Clustering

To help visualise the occupancy dynamics of Pol II and ER-α at enhancers and genes we clustered the data with the R-implementation of Affinity Propagation (AP) (*Frey & Dueck, 2007*). AP is a clustering method based on belief propagation and works iteratively by passing messages between data points until exemplars (cluster centres) automatically emerge. A preference parameter $p$ has an effect on the final number of clusters. The R implementation of AP can search through values of $p$ to achieve an approximately pre-specified number of clusters. The method is similar to k-means but can achieve much better optimisation of the k-means objective function than the standard EM algorithm.

To reduce the effect of noise, for Pol II we clustered only the pairs of the time series for which the Pearson correlation coefficient was at least 0.2 between replicates and the total number of mapped reads was at least 30. For ER-$\alpha$, due to lack of replicates, we only clustered the time series with more than 100 reads in total across all times. Prior to the clustering we standardized each time series to $z$-scores to bring all time series onto the same scale. We obtained 20 and 22 clusters for Pol II time series over enhancer and genes, respectively. Similarly we obtained 21 and 21 clusters for ER-$\alpha$ time series over enhancer and genes. We also jointly clustered time series of PolII and ER-$\alpha$. The results of the clustering can be seen in Fig. S2.

## Enhancer-centric model

Suppose that an enhancer $j = 1,\ldots,J$ regulates a gene $k = 1,\ldots,K$ at a number of time points, and that their contact is mediated by a protein. We can expect that the time course data of ChIP-seq data at an enhancer $j$ i.e., $\boldsymbol{X}_j = (x_{j,1},\ldots,x_{j,D})$ and gene $k$ i.e., $\boldsymbol{Y}_k = (y_{k,1},\ldots,y_{k,D})$ would on average be more correlated for interacting pairs than their non-interacting counterparts. Here, we intend to learn the underlying distribution of correlations of the two classes of pairs for four complementary datasets and on their basis jointly classify a new unobserved instance. In addition, we combine the time course derived attributes with the corresponding distribution of genomic separation for interacting and non-interacting elements.

## Definition of the model

Our model is defined in terms of two $K$-dimensional random variables $\boldsymbol{I}_j = I_{j,1},\ldots,I_{j,K}$ and $\boldsymbol{D}_j = D_{j,1},\ldots,D_{j,K}$. The first variable $\boldsymbol{I}_j$ encodes a structure of simultaneous contacts of a given enhancer $j$ with its surrounding $K$ putative target genes. It has $K$ binary entries $I_{j,k}$ indicating whether $(E_j, G_k)$ forms an interacting ($I_{j,k} = 1$) or non-interacting pair ($I_{j,k} = 0$). The variable $\boldsymbol{D}_j$ is a $K \times N$-dimensional matrix of observed attributes with each row $(D_{j,k})$ consisting of $N$ values of pair-wise comparisons between time series of an enhancer $j$ and a gene $k$, and their genomic location. The first set of comparisons rely on Pearson correlation and involves calculating its value $c_{j,k,n}$ for each pair $(E_j, G_k)$, i.e., its time series $(\boldsymbol{X}_{j,n}, \boldsymbol{Y}_{k,n})$, and for each dataset $n \in N$, where $N$ is a number of time course ChIP-seq datasets. Additionally, the data vector also contains the Euclidean distance $d_{j,k}$ calculated between the genomic coordinates of the canonical TSS of a gene $k$ to the centre of an enhancer $j$.

The joint likelihood of the model can be written as:

$$P(\boldsymbol{D}_j, \boldsymbol{I}_j) = P(\boldsymbol{D}_j | \boldsymbol{I}_j) P(\boldsymbol{I}_j). \tag{1}$$

The model provides a probability of observing a particular $\boldsymbol{D}_j$ under a given structure $\boldsymbol{I}_j$. Due to its regulatory role, an enhancer is unlikely to regulate a high number of genes, thus we can expect that the true $P(\boldsymbol{I}_j)$, which in the Bayesian treatment is a prior distribution over the structures, would be sparse. Moreover, we could expect that $D_{j,k}$ and $D_{j,k'}$ of any two interacting pairs $k, k'$ would be interlinked, as correlations between gene-enhancer pairs are not independent variables. These dependencies would be reflected in a true

form of the likelihood $P(\mathbf{D}_j|\mathbf{I}_j)$. Lastly, we could also expect that the $N+1$ attributes i.e correlations $c_{j,k,n}$ and distance $d_{j,k}$ of a pair $j,k$ of the vector $D_{j,k}$ would also be correlated.

## Simplifying the likelihood and Naive Bayes

The modelling of all dependencies however is difficult given the relative sparsity of our training data. We therefore restrict the form of the joint distribution and construct an approximate joint probability of enhancer-gene contacts. Pairwise correlations provide a valid likelihood if we restrict our model to consider one gene per enhancer.

### (a) The joint distribution factorises

We assume that the likelihood $P(\mathbf{D}_j|\mathbf{I}_j)$ can be factorised and written in the form:

$$P(\mathbf{D}_j|\mathbf{I}_j) = \prod_{\{k:I_{j,k}=1\}} P(D_{j,k}|I_{j,k}=1) \prod_{\{k:I_{j,k}=0\}} P(D_{j,k}|I_{j,k}=0) \tag{2}$$

where $\mathbf{I}_j = I_{j,1},\ldots,I_{j,K}$ and $\mathbf{D}_j = D_{j,1},\ldots,D_{j,K}$. Hence the distribution of each $D_{j,k}$ is conditionally independent of other allocations and conditional only on the indicator variable $I_{j,k}$.

### (b) An enhancer regulates a single gene

We assume further, that an enhancer $j$ can interact with only one gene $k$. We restrict the event space of $P(D_j,I_j)$ to its subspace $P(D_j,I_{j,k}^{(1)})$, where $I_{j,k}^{(1)} = 0,\ldots,\underset{k\text{th}}{1},\ldots,0$ . From (2) the events are given by:

$$P(\mathbf{D}_j|I_{j,k}^{(1)} = 0,\ldots,\underset{k\text{th}}{1},\ldots,0) = P(D_{j,k}|I_{j,k}=1) \prod_{\{l:l\neq k\}} P(D_{j,l}|I_{j,l}=0). \tag{3}$$

The prior distribution $P(\mathbf{I}_j)$ follows a multivariate Bernoulli distribution, and thus the restriction is equivalent to setting the probabilities of all the structures $\mathbf{I}_j$ with non-singular number of contacts i.e., $\mathbf{I}_j^{(2)},\mathbf{I}_j^{(3)},\ldots,\mathbf{I}_j^{(K)}$ to zero. For the remaining $\mathbf{I}_{j,k}^{(1)}$ we assume that the prior is uniform across these sparse vectors, i.e.,

$$P(\mathbf{I}_{j,k}^{(1)} = 0,\ldots,\underset{k\text{th}}{1},\ldots,0) = 1/K , \tag{4}$$

so that each $\mathbf{I}_{j,k}^{(1)}$ is equally likely *a priori*.

### (c) The distribution of attributes is independent

Assuming that the attributes are conditionally independent, the likelihood component $P(D_{j,k}|I_{j,k})$ becomes:

$$P(D_{j,k}|I_{j,k}) = P(d_{j,k},c_{j,k,1},\ldots,c_{j,k,N}|I_{j,k}) = P(d_{j,k}|I_{j,k}) \prod_{n\in N} P(c_{j,k,n}|I_{j,k}) \tag{5}$$

where $d_{j,k}$ is a distance from the centre of an enhancer $j$ to the TSS of a gene $k$, whereas $c_{j,k,n}$ is a correlation between the time series of the $n$th time course dataset between an enhancer $j$ and gene $k$.

Combining the assumption of the factorisable likelihood (2) with the conditional independence of attributes (5) yields,

$$P(\mathbf{D}_j|\mathbf{I}_j) = \prod_{k=1}^{K} P(D_{j,k}|I_{j,k}) = \prod_{k=1}^{K} \left[ P(d_{j,k}|I_{j,k}) \prod_{n\in N} P(c_{j,k,n}|I_{j,k}) \right]. \tag{6}$$

Restricting the event space to single enhancer-gene events (3) results in,

$$P(\boldsymbol{D}_j|\boldsymbol{I}_{j,k}^{(1)}) = \left[ P(d_{j,k}|I_{j,k}=1) \prod_{n\in N} P(c_{j,k,n}|I_{j,k}=1) \right]$$
$$\times \prod_{\{l:l\neq k\}} \left[ P(d_{j,l}|I_{j,l}=0) \prod_{n\in N} P(c_{j,l,n}|I_{j,l}=0) \right]. \tag{7}$$

The assumption of conditional independence of features in (5) and the fact that each vector $\boldsymbol{I}_{j,k}^{(1)}$ is a 1-of-K (i.e., one-to-one relation) representation of $K$ class indicators makes this algorithm a special case of Naive Bayes (NB) model.

## Posterior

The posterior distribution under the model is:

$$P(\boldsymbol{I}_{j,k}^{(1)}|\boldsymbol{D}_j) = \frac{P(\boldsymbol{D}_j|\boldsymbol{I}_{j,k}^{(1)})P(\boldsymbol{I}_{j,k}^{(1)})}{\sum_{k=1}^{K} P(\boldsymbol{D}_j|\boldsymbol{I}_{j,k}^{(1)})P(\boldsymbol{I}_{j,k}^{(1)})}. \tag{8}$$

The posterior distribution can be used to find the probability of each structure $\boldsymbol{I}_{j,k}^{(1)}$ given the pair-wise comparisons in $\boldsymbol{D}_j$, i.e., the values of the data-specific correlations and distance for each pair $(E_j, G_k)$ and all complementary pairs $(E_j, G_{\{l:l\neq k\}})$. The posterior probabilities can be used to infer the most likely target of an enhancer $j$ out of $K$ genes.

## Positive set of interactions and background negatives

We overlap the distal enhancers and promoter-extended-genes with the combined set of ChIA-PET predicted links using both ER-$\alpha$ and Pol II antibodies from ENCODE/GIS-Ruan (*Li et al., 2012*) (GEO accession numbers GSM970209 and GSM970212). The overall design and processing of the datasets is described under GEO accession number GSE39495. The sources contain the high-confidence binding sites and protein-mediated chromatin interactions with three and four replicates for ChIA-PET with antibodies for ER-$\alpha$ and Pol II respectively. Overlapping the enhancers and genes with the concatenated set of empirically confirmed interactions revealed a total of 2,733 enhancer-promoter links, and shows that 2,087 of our distal enhancers interact with at least one promoter.

To define the negative set, we restricted ourselves to all enhancer-gene pairs involving known interacting enhancers coming from the positive set and all the remaining non-targeted genes. Enhancers without any confirmed interactions from ChiA-PET data were not used for training as we have no information about their target genes.

## Data features and their distributions

The method uses five features of two types, i.e., four correlations and one distance. To obtain the first four we correlated ChIP-seq time series at enhancers with those at promoter-extended genes, for each dataset, for all enhancer-gene pairs in the positive and negative set (as defined above). For Pol II we used the average correlation across the two replicates. For the distance feature we used the $\log_{10}$ of genomic distance between the centre of the enhancer and the canonical TSS of an extended gene. We used the training set to estimate the distributions $P(c_{j,k,n}|I_{j,k})$ and $P(d_{j,k}|I_{j,k})$ using kernel density estimation

(KDE) with a Gaussian kernel. To ensure that the bandwidths of positive distributions are biologically meaningful and robust, we used cross-validation. As part of the approach, we sequentially removed all features of each chromosome from their total set across all chromosomes and at each time calculated the log-likelihood of KDE for the reduced set of features. We then used the value of the bandwidth with the highest log-likelihood over left-out data. In contrast, due to a large number of negative examples and computational cost associated with KDE, employing the same approach for negatives was infeasible. Their size, however, also entails less requirement for optimised fitting, and thus to select the bandwidth we resorted to the Scott's rule (*Scott, 2015*).

## Model validation

We trained the classifier on the odd chromosomes and estimated the training error. Similarly, we tested the method on the even chromosomes and obtained the test error. Since the test data is not used to build the classifier (i.e., fit the feature densities), its predictions on the test data can be considered unbiased. We measured the performance in two ways. Firstly, we evaluated and plotted precisions against the True Positive Rate (TPR or recall) of 10%, 20%, and 30% for various combinations of features. Secondly, we used an alternative MAP measure. Under our model each enhancer possesses a maximum a posteriori (MAP) gene which is our best guess of enhancer's target. The MAP measure is the percentage of times the MAP inferred target gene is confirmed by the positive set of interactions in the ChIA-PET data.

## Performance within and outside TADs

We stratified our predicted interactions at 10%, 20%, and 30% thresholds into those that lie within domains and those that crossed domain boundaries. Each TPR threshold maps to a subsets of negative and positive links, and therefore each subset was partitioned into inter- and intra- domain interactions. We then tested precisions for each of the subsets. For details of TAD preparation refer to the Supplementary Material (suppl: Domains conserved between mESC, mouse Cortex, hESC and IMR90 converted from hg18 to hg19 using the NCBI Genome Remapping Service; http://www.ncbi.nlm.nih.gov/genome/tools/remap)

## Prediction of target genes

We used our model to infer gene targets with strong evidence of being regulated by at least one enhancer. The probability of gene $k$ having at least one active regulatory link from an enhancer under our model is defined,

$$P(card(\{j \in J : I_{j,k} = 1\}) > 0) = 1 - \prod_{\{j \in J : I_{j,k} = 1\}} (1 - P(I_{j,k}^{(1)} | \boldsymbol{D}_j)) \quad (9)$$

where the product above is equal to the probability that no enhancers regulate the gene.

*Hah et al. (2011)* carried out GRO-Seq experiments (GEO accession number GSM678536) to detect whether Pol II molecules are engaged in transcription at the start of the experiment. The experiments were performed with the same cell-line and stimulation as ours and were used to determine the early transcriptional response of genes following E2 treatment. Using these data and the regulation probability scores defined in

Eq. (9), we assessed how many of our predicted distally regulated genes were differentially expressed at early time points. Using the EdgeR processed GRO-seq data we filtered the GRO-seq determined DE genes at 10, 40, 160 min after E2 stimulation with $q$-value (multiple hypotheses testing adjusted $p$-values from EdgeR) of less than 0.05, 0.01, 0.001. For each $q$-value, we combined the DE genes from each of the time points into a single list.

## RESULTS AND DISCUSSION

We demonstrate our method using ChIP-Seq time course data collected from the MCF7 breast cancer cell-line stimulated by estrogen. After stimulation, the ER-$\alpha$ TF associates with numerous enhancers to regulate transcription of target genes. ER-$\alpha$, encoded by the ESR1 gene, is a particularly well studied example of a nuclear receptor due to its role in breast cancer development. Its genome-wide binding pattern under stimulation with estrogen has been established through ChIP-seq experiments (*Liu & Cheung, 2014*; *Magnani & Lupien, 2014*; *Ross-Innes et al., 2012*). Here, the genome-wide occupancy of ER-$\alpha$ along with RNA polymerase (Pol II) and two histone marks (H3K4me3 and H2AZ) associated with transcriptional competence, were measured via ChIP-seq at eight consecutive time-points after exposure of cells in estrogen free media to estradiol. ChIA-PET data are also available in this system and were used to evaluate our method's performance (*Fullwood et al., 2009*; *Li et al., 2010*; *Li et al., 2012*).

### ER-$\alpha$ bound enhancers overlap experimentally determined promoter interaction regions

To locate binding events formed after stimulation with estradiol, we determined a set of genomic loci associated with ER-$\alpha$ in at least two time points. Among these 47,921 regions, 21,336 overlapped with a known gene or within a 300 bp region upstream from its TSS (promoter-extended gene region) while 26,585 were distant from genes (distal enhancers).

Next, we determined how many of our distal ER-$\alpha$-bound enhancers are known to form links with promoter-extended genes. Overlapping regions with interactions derived from two public ChIA-PET datasets that used the same ER-$\alpha$ and Pol II antibodies revealed a total of 2,733 enhancer-promoter links. These interactions were used as a positive set for the purpose of developing our classifier. Missing interactions involving the same enhancers and other promoters in the same chromosome were used as the negative set. When training and testing the classifier, we did not include enhancers that did not have any interactions according to the ChIA-PET data. These enhancers are most likely not detected by the ChIA-PET method due to its limited sensitivity and their inclusion would introduce many false negatives into our training and testing data. However, we apply the classifier to all enhancers when making target gene predictions.

### ChIP-seq time series data

We calculated the number of mapped reads for each of our ChIP-seq datasets over promoter-extended-gene bodies and over our consensus ER-$\alpha$ binding sites to create time series data for genes and enhancers (see 'Materials and Methods'). We clustered the ER-$\alpha$ and Pol II data to help visualise the occupancy dynamics at enhancers and genes. As shown

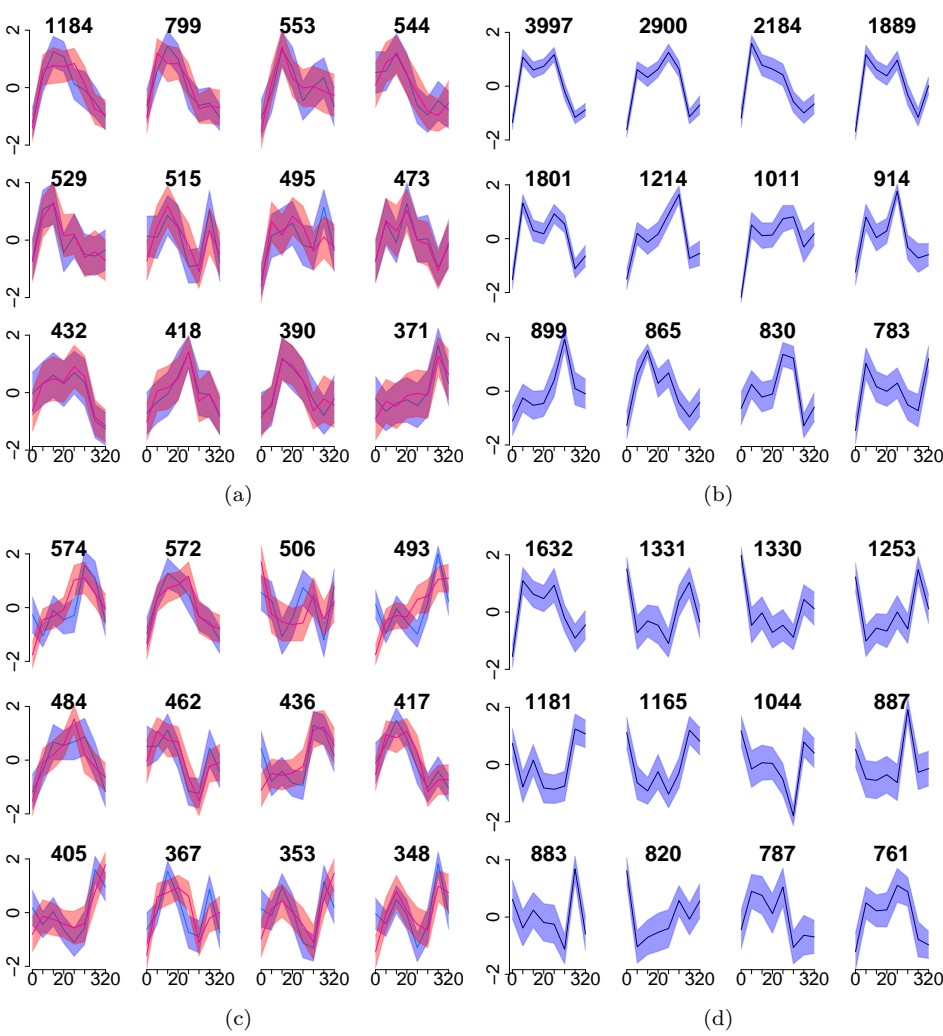

**Figure 1** **ChIP-seq time course data show a variety of dynamic profiles which are exploited by our classifier.** (A, C) show profiles of the first (blue) and the second (magenta) replicate of Pol-II for enhancers and genes, respecively. (B, D) show profiles of ER-$\alpha$ for enhancers and genes, respectively. $X$-axis shows time, $Y$-axis shows $+/-$ one standard deviation of $z$-scores in each cluster. The headers show the number of time series in each cluster.

in Fig. 1, the clusters show substantial differences in occupancy dynamics across both genes and enhancers. This is expected for Pol II which shows a broad range of response profiles in this system (*Honkela et al., 2015*). Additionally, some differences in ER-$\alpha$ profiles were also detected, suggesting that occupancy is not solely determined by the nuclear concentration of ER-$\alpha$.

## Time series correlation and distance-based features are informative about enhancer-promoter interactions

We calculated the Pearson correlation coefficient between enhancer and gene time series data for every enhancer-promoter pair in the positive and negative set. Figure 2 shows the distribution of correlations for each dataset in our training data (odd chromosomes).

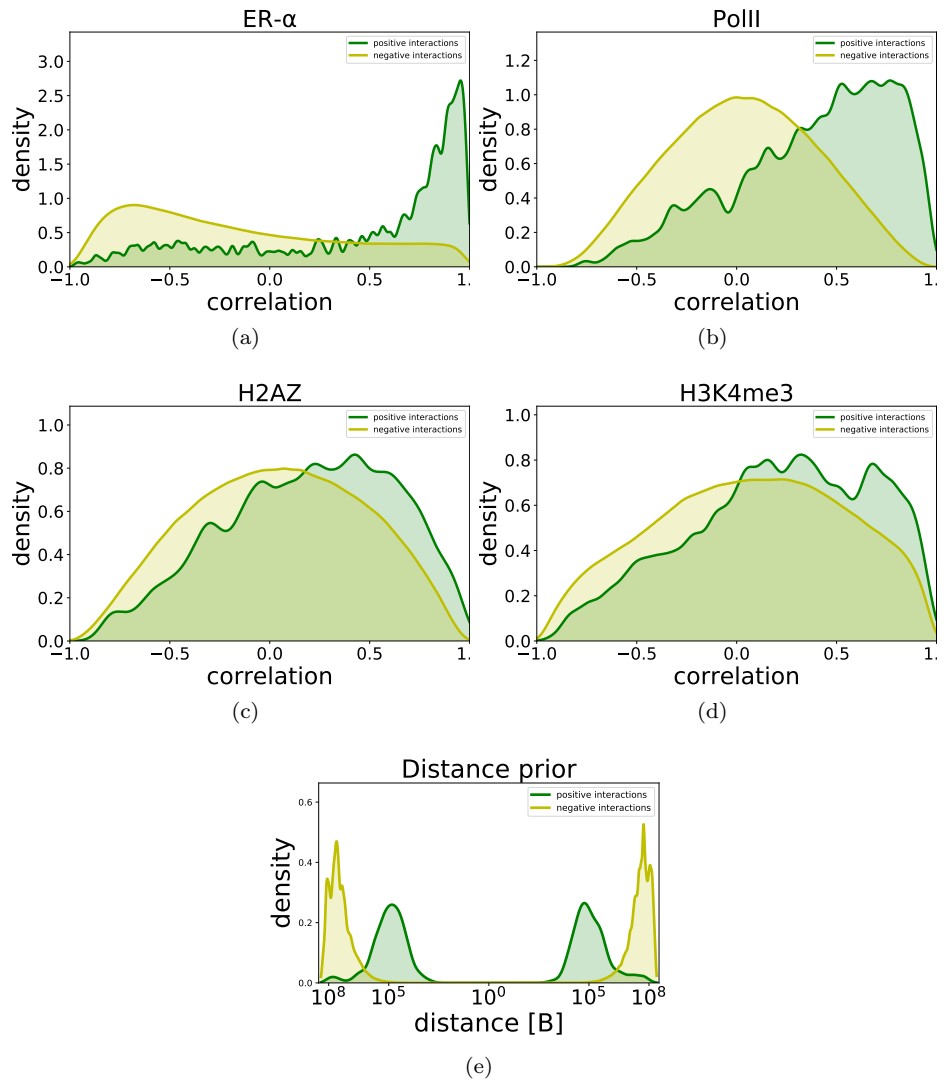

**Figure 2** **Distribution of correlation of time series data (A–D) and genomic distance (E) for promoter-enhancer pairs and for non-interacting pairs.** Here we define positive links as those confirmed by ChIA-PET experiments while negative links are defined as those not supported by ChIA-PET and involving the same set of enhancers. We observe that positive links tend to have higher correlations in the ChIP-Seq data compared to negative links, with the effect strongest for ER$\alpha$ and Pol-II.

The distribution for positive interactions differs substantially from the background for all four datasets, with interacting regions more highly correlated on average. This difference is most pronounced for ER-$\alpha$ and Pol II (Figs. 2A and 2B) while there is a much smaller difference for the histone marks H2AZ and H3K4me3 (Figs. 2C and 2D). We also compare the distribution of genomic separation for interacting and non-interacting promoters and enhancers in Fig. 2E. Although a highly informative feature, there is a substantial overlap in the positive and background distance densities due to a large separation of many ER-$\alpha$ bound enhancers from their target promoters; therefore, distance alone is insufficient for accurate prediction of interactions. We note that our ChIA-PET data does not contain

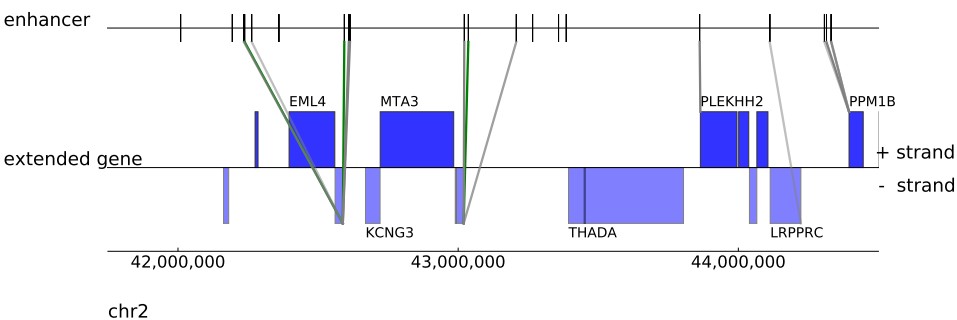

**Figure 3** An example of predictios with posterior probabilities above cut-off thresholds with FDR of 20%, 25%, 30% (**indicated by different shades of green/gray**). The green/grey colour of each link indicates whether the prediction is confirmed/unconfirmed by the ChIA-PET data.

very short ChIA-PET links. Links of a size shorter than 4.5 kB are usually considered to be the result of self-ligations and are filtered out *Li et al. (2010)*. In Fig. S3 we plotted the corresponding histograms using data from all chromosomes. We observe that the distribution does not change with the addition of data from even chromosomes.

## Naive Bayes classifier performance

We developed a Naive Bayes classifier which integrates several discriminative features to estimate the probability of interactions between enhancer and putative target genes. Figure 3 shows predicted interactions with only a small number confirmed by ChIA-PET (green). Interactions are shown using different shading for classification probabilities above 0.72, 0.54, 0.49 thresholds corresponding to 0.2, 0.25, 0.3 FDR levels (posterior probabilities with the highest TPR which are associated with the selected FDRs (1-precision)) estimated using the training data (combination of features: Pol II, ER, distance).

We evaluated classifier performance using precision-recall (PR) curves (Figs. 4A–E and 4F). The classifier was trained on data from odd chromosomes and the results were used to establish which combination of features is most informative. Data from even chromosomes was then used as an unbiased test set to establish the performance of the selected model and to estimate decision cut-off levels. However, we do not observe significant over-fitting, probably due to the small number of features used by the classifier. Comparison of different combinations of correlations and distance features, including distance-alone and correlation-alone variants, shows that data from ER-$\alpha$ can be combined with distance to greatly enhance predictive performance (results for all possible feature combinations are shown in the Supplemental Information) while data from Pol II provides a smaller improvement in performance. The H2AZ and H3K4me3 time course data were found to not be particularly informative, consistent with Fig. 2 which shows these histone marks to have a less pronounced difference in distribution for positive and negative links. Table 1 shows that using the probability cut-offs to infer links across 23 chromosomes our model (combination of features: PolII, ER, distance) consistently outperforms the distance-alone model in terms of the number of uncovered true links. We show that at FDR equal to 0.20 our model infers 26.7 times more interactions than predictions based on proximity alone (see Table 1). In addition to considering precision-recall curves, we also tested how often

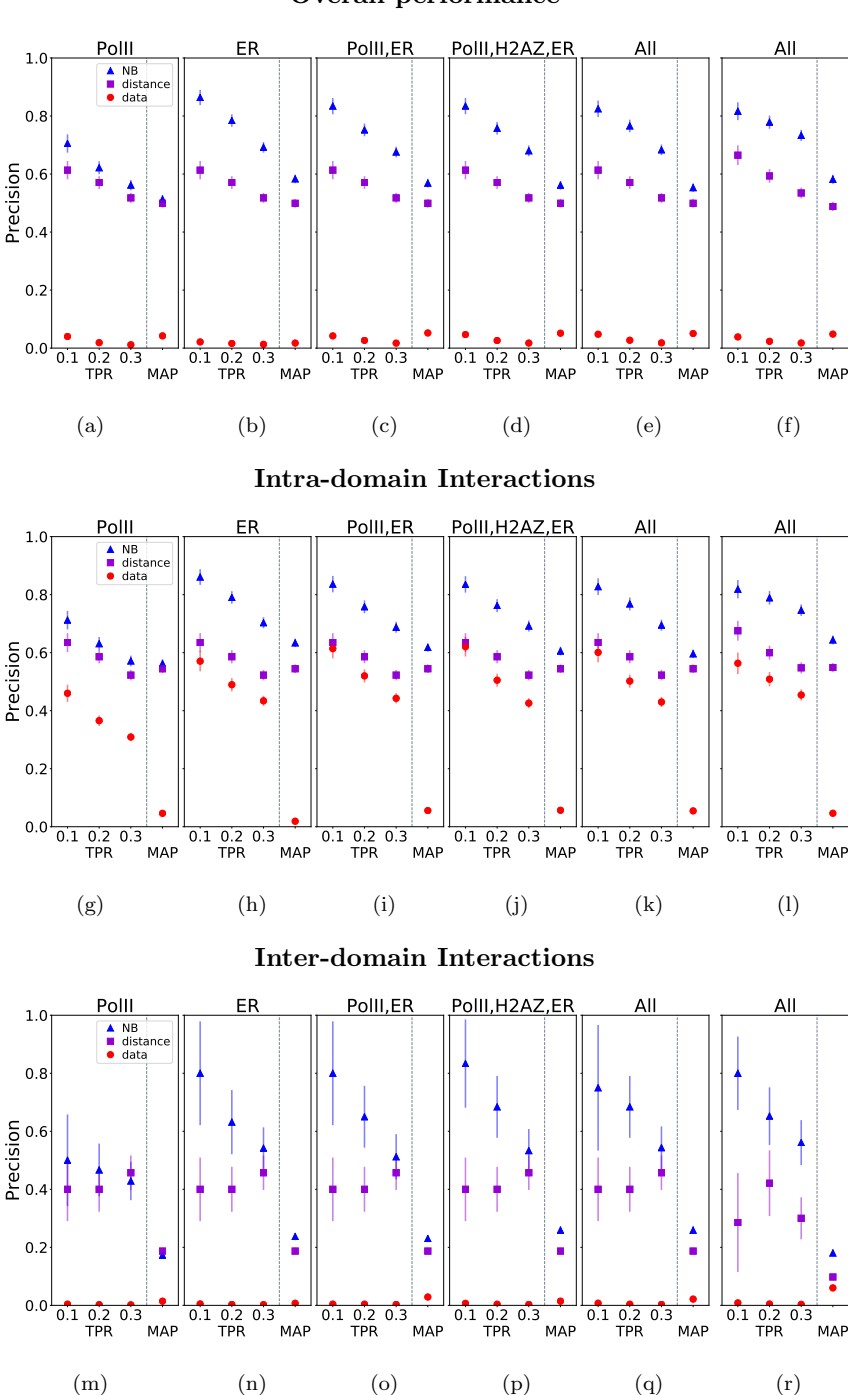

**Figure 4** **Graphs (A–R) show the performance of the model, measured by Precision-Recall and MAP scores.** The precisions are plotted againsts TPR of 0.1, 0.2, 0.3. Each column shows the performance of the model with a variant of correlation-based feature/s (i.e., data, see header) and proximity-based feature (i.e., distance, see header). The first five columns of each row show the performance on the training data. The last column shows the performance on the test data.

**Table 1** True links uncovered at decreasing false discovery rates for distance alone and distance assisted models.

| FDR | Data/distance | Distance | Ratio |
|---|---|---|---|
| 0.4 | 14,217 | 6,041 | 2.4 |
| 0.3 | 7,531 | 1,124 | 6.7 |
| 0.2 | 2,800 | 105 | 26.7 |
| 0.1 | 109 | 49 | 2.2 |

using maximum a posteriori probabilities (MAP) to link all enhancers (in the training and test data) to their most probable promoters would result in correct assignments according to the ChIA-PET data (right-most column of plots in Figs. 4A–E and 4F). The mean performance in the MAP case is reduced and the added value of the ChIP-Seq data relative to the proximity information is also reduced. This is because for many enhancers the ChIP-Seq data signal is relatively weak and therefore focussing on the enhancer-promoter pairs with higher classification probabilities (as in the PR curves approach) produces better quality prediction on average than when we make predictions for all enhancers.

## Inter-domain and intra-domain predictions

Most enhancer-promoter interactions are thought to occur within the same Topologically Associating Domain (TAD) and we were interested in whether our method can discover interactions across TAD boundaries. In order to assess the performance of the model on discovery of intra-domain interactions and the ones involving elements from two different domains, we stratified our predicted interactions into those two groups, and recomputed precision-recall and MAP performance (Figs. 4G–K/L–4M–Q/R).

The majority (79%) of enhancer-promoter interactions lie within domains. The PR curves in Figs. 4G–K and 4L show that the ER-$\alpha$ and distance features provide the greatest contribution to performance. The Pol-II feature is also informative but does not add much to performance when combined with the ER-$\alpha$ data. Interestingly, within domains the "data-alone" model possesses much higher predictive power than in the chromosome-wide model. By excluding the possibility of long-range interactions beyond domain boundaries, the number of false positives is greatly reduced. Nevertheless, we see that incorporation of the distance feature still improves classification performance within domains.

On the contrary (see Figs. 4M–Q and 4R) focusing on the remaining inter-domain interactions we notice that, in consequence of a large number of negative interactions, the correlation data alone is insufficient for classification. The proximity data, despite being much better than the data-alone, also does not offer the performance that we achieved for the intra-domain cases. However, distance-assisted models perform much better than data-alone and distance-alone models and the top-ranked links have similar precision than in the intra-domain case. Note however that the MAP results are much lower for the inter-domain predictions, suggesting that many enhancers linking to promoters across TAD boundaries according to the ChIA-PET data do not have this as their top-scoring interaction according to the model.

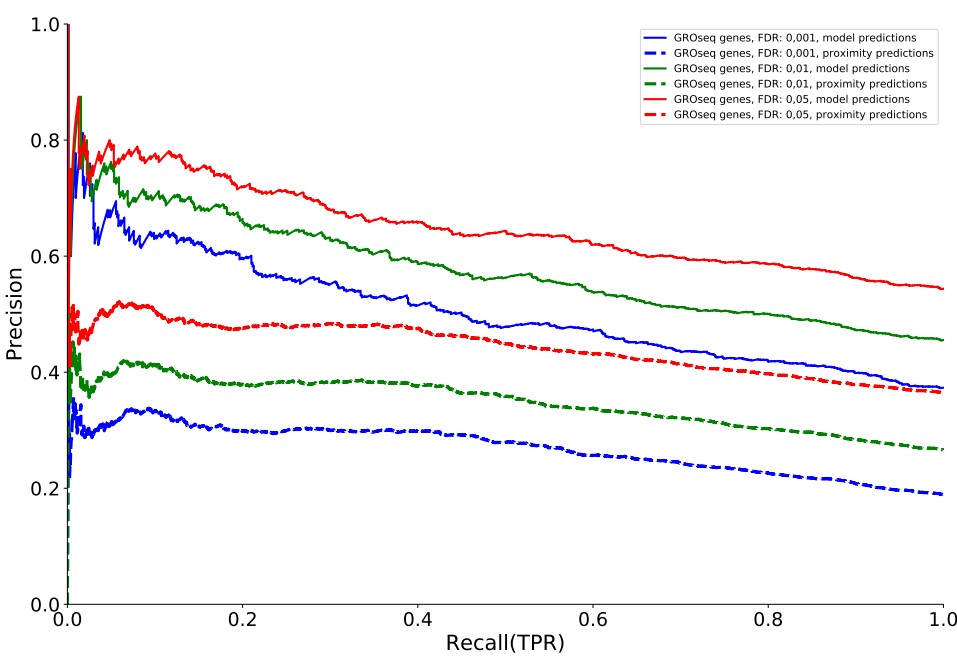

**Figure 5** The Precision-Recall curves assess the model on the ability to predict differentially expressed genes (as derived from GRO-Seq data), given a number of model-assigned regulators of each gene and the confidence of each prediction.

## Testing alternative dataset design choices

Our selection of data features involved some arbitrary choices and therefore we considered robustness to varying some of the parameters used. We first investigated alternative promoter region sizes for promoter-gene regions, their effect on test and training sets and the effect on the performance of the model. The comparison between the distributions of features in Figs. 2 and S4 and between PR curves in Figs. S5–S8 show that increasing the promoter size up to 1,500 bp upstream from a gene causes neither no changes to the distributions of features nor to the overall performance, and thus the model is robust to changes in promoter region size. Similarly, Figs. S9 and S10 show that using alternative parametrisation of MACS in which we switched on $\lambda_{local}$ parameter produces similar results to our default parametrisation where we switched that parameter off. Figure S11 shows that the distributions of features remain similarly unchanged.

We have used eight timepoints for this study, but most epigenomic time course datasets from a single stimulation have fewer available timepoints. We therefore assessed the performance for reduced datasets with six and four timepoints in Fig. S12. Inclusion of data with less timepoints reduced the performance of the data-only method substantially, but combining data with the prior still leads to a significant improvement over the prior-only model even with only four timepoints.

## Validation of ER-regulated target gene predictions

Finally, we used our method to provide a highly confident (FDR of 0.25) list of directly ER-regulated target genes in this system. This list (Table S1) includes 1,978 genes with at least one predicted enhancer link.

In Fig. 5 we compared our set of predicted distally regulated genes against a list of early differentially expressed genes obtained from GRO-seq experiments (Hah et al., 2011). PR curves showed that the larger the value of the score (see Materials and Methods), which is roughly proportional to the number of times a gene is predicted to be a target of distal enhancer, the higher the chance that the gene is differentially expressed. Using a score based only on proximity of ER-$\alpha$ binding events is much less predictive of early differential expression.

## CONCLUSIONS

We have developed a Bayesian method which is capable of integrating genomic distance with a correlation of ChIP-seq time series in order to predict physical interactions between enhancers and promoters. We evaluated the performance of our method against ChIA-PET predicted links and using different combinations of features. Using complementary GRO-seq data from the same cell-line and stimulation we show that our model can accurately predict distally regulated, differentially expressed genes under stimulation with estrogen.

Experimental approaches to identifying ehancer-promoter interactions genome-wide are increasingly popular but have some limitations. ChIA-PET datasets typically only identify a relatively small number of enhancer-promoter interactions with confidence, while HiC data typically has too low genomic resolution to resolve specific enhancer interactions. Even the more focussed Capture-HiC protocals are limited to restriction fragments of several kb and HiC data are generally associated with complex noise characteristics requiring sophisticated corrections for background. Our model can therefore serve as a useful complementary approach to these techniques and offers insight into stimulation-dependent, and cell-type specific transcriptional regulation.

In this work we have focussed on intergenic enhancers, because our data contains Pol-II ChIP-Seq data which has transcriptional signal on introns and is therefore not ideally suited for identifying intronic enhancers. However, the computational method could potentially work for intronic enhancers with different ChIP-Seq data combinations. For example, with access to enhancer enriched epigenomic marks such as H3K27ac or H3K4me1 then the data may be suitable for identifying links involving intronic enhancers.

### Funding

The work was funded by the European ERASysBio+ Initiative Project Systems Approach to Gene Regulation Biology Through Nuclear Receptors (SYNERGY) (Biotechnology and Biological Sciences Research Council Grant BB/I004769/2 (to Magnus Rattray), Academy of Finland Grant 135311 (to Antti Honkela), and Bundesministerium fur Bildung und

Forschung Grants ERASysBio+ P#134 (to George Reid) and 0315715B (to KG). Magnus Rattray and KG were further supported by the European Union Seventh Framework Programme Project RADIANT (Rapid Development and Distribution of Statistical Tools for High-Throughput Sequencing Data) (Grant 305626), and Antti Honkela was further supported by Academy of Finland Grants 259440 and 284642. Also, Magnus Rattray and Mudassar Iqbal received the MRC award MR/M012174/1. The funders had no role in study design, data collection and analysis, decision to publish, or preparation of the manuscript.

### Grant Disclosures

The following grant information was disclosed by the authors:
Biotechnology and Biological Sciences Research Council: BB/I004769/2.
Academy of Finland: 135311.
Bundesministerium fur Bildung und Forschung: ERASysBio+ P#134, 0315715B.
European Union Seventh Framework Programme Project RADIANT: 305626.
Academy of Finland: 259440, 284642.
MRC award: MR/M012174/1.

### Competing Interests

The authors declare there are no competing interests. Korbinain Grote is employed by Genomatix Software GmbH.

### Author Contributions

- Tomasz Dzida analyzed the data, contributed reagents/materials/analysis tools, wrote the paper, prepared figures and/or tables, reviewed drafts of the paper, wrote code implementing the computational method.
- Mudassar Iqbal analyzed the data, prepared figures and/or tables, reviewed drafts of the paper.
- Iryna Charapitsa and Filomena Matarese performed the experiments, reviewed drafts of the paper.
- George Reid, Antti Honkela and Henk Stunnenberg conceived and designed the experiments, reviewed drafts of the paper.
- Korbinian Grote analyzed the data, mapped the reads, supplied text for GEO submission.
- Magnus Rattray conceived and designed the experiments, wrote the paper, reviewed drafts of the paper.

### Data Availability

Code: https://github.com/ManchesterBioinference/EP_Bayes.
Data available from GEO: accession GSM2467201
https://www.ncbi.nlm.nih.gov/geo/query/acc.cgi?acc=GSM2467201.

### Supplemental Information

Supplemental information for this article can be found online at http://dx.doi.org/10.7717/peerj.3742#supplemental-information.

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
