# Peer review of "Predicting stimulation-dependent enhancer-promoter interactions from ChIP-Seq time course data"

_PeerJ, doi:10.7717/peerj.3742_

## Round 0.1 · original submission · Major Revisions

Your manuscript has been reviewed by three experts in the field. As you will find from their comments below, all of them are basically interested in your study but also raise a number of criticisms. Please read these comments carefully and revise the manuscript accordingly. Though most of these comments are minor, note that all of them point out the problems in presenting figures and their legends, in particular.

Reviewer 1 ·

Basic reporting

I think that in general the article is well written, in clear professional English. Literature is in general cited sufficiently (with one exception described below). However, some details are missing and especially the figures need improvement.

Major points:

My main point of criticism here is about the quality of the figures and the lack of explanation about what they are showing. The reader basically has to make too many assumptions to be able to interpret them with confidence.

- Fig. 1: what are the ranges shown? Center of the cluster +/- SD? What are the numbers above each plot? The number of regions in the cluster? What is the Y axis? Log fold change? Why are 12 clusters shown in each subplot? Another problem is that many clusters seem practically indistinguishable from each other (see for example the four top plots in Fig. 1a). Should these really be considered as different clusters? They might be merged, I feel.
- For Fig. 2e, people will wonder why the density is 0 for the smaller distances. I guess this is because of the way intergenic enhancers were defined here. It would be better to indicate this in the figure; for example: mark the region (+/-300bps?) where enhancers would be filtered out.
- For Fig. 3: the colors in Fig. 3a are not clear. Adding gene symbols to the genes might help the understanding. Fig. 3b-g: Adding some distance between the training and test panels would make things more intuitive. An explanation is needed about what “PolII”, “ER”, etc on top of each plot means, as well as what “NB” stands for, and what is the meaning of “distance” and “data”. Please also explain what the error bar means. At present, all of this has to be guessed by the reader. There appears to be a “3” in Fig. 3f, third panel, which should be removed. Finally, at present 5 panels are shown for training data, and only 1 for test data. It would make more sense to show performance on the test data in more detail, rather than on the training data, as this performance gives a better idea of the general performance on new data.

Minor points:

- In November 2016, several papers (see Cell Volume 167 Issue 5, including Spurrell et al., Cell, 2016; Javierre et al., Cell, 2016; etc) were published by the BLUEPRINT Consortium/ International Human Epigenome Consortium (IHEC) describing analysis of promoter-enhancer interactions using promoter capture Hi-C in multiple cell types and conditions. I think a reference to these studies and their expected impact in the future should be added to the Introduction.
- I think that no reference was given for MACS. This should be added.
- p4 line 157. “The method is similar to k-means but can achieve much better optimisation of the k-means objective function than the standard EM algorithm.” As far as I understand k-means and the EM algorithm are two different things, and k-means does not use EM. I would recommend removing this sentence, or providing a citation that supports this statement.
- P8 line 320: what are “0.66, 0.47, and 0.35”? Is this correlation? A more clear explanation is necessary.

Experimental design

I think that the paper described original research, with a well defined and clear problem and aim. The computational methodology is described in sufficient detail and is quite interesting.

However, there are some issues about the newly produced ChIP-seq data (see below).

Major point:
- New ChIP-seq data (ER-alpha, H3K4me3, and H2AZ) was produced, but details about the experimental procedures are completely missing from the Materials and Methods section. How were antibodies obtained? How about sequencing libraries, buffer solutions, concentration, cell counts, etc?

Raw data appears to have been submitted to NCBI GEO, however, the data is still set to private, and no reviewer URL was provided, so I can not comment on the data itself.

Minor points:

- Were control samples produced for the ChIP-seq data? If no, some discussion about how this could influence the downstream analysis would be useful. Differences in chromatin accessibility can result in non-specific peaks which are hard to remove without using control samples. Some ChIP-seq peaks might therefore simply reflect open chromatin structure, and promoter-enhancer correlations could reflect similar changes in chromatin accessibility, rather than purely binding by ER-alpha or Pol2-II.
- In the Materials and Methods section (p5 line 201~) the assumption is made that each enhancers regulates a single gene. I think that this is in practice a reasonable assumption, as it becomes computationally impossible to consider more combinations of enhancer-to-promoter interactions. However, it would seem quite reasonable and doable to add the case where an enhancer does not regulate any of the genes to the model. It would be quite interesting to see what happens if this case (“I(0)_j”?) is included. Does it change the assignments? Will you end up with many enhancers having no apparent target promoter?

Validity of the findings

In my opinion the findings are valid, though they are not so easy to interpret with confidence from the figures (see comments above). Improving the quality of the figures are their explanation will improve this point.

Additional comments

Some additional (brief) discussion of the approach and results might be of interest for people working on related topics.
- Currently you are considering only the correlation between consistent features (for example: ER-alpha at the promoter vs ER-alpha at the enhancer). However, it might be possible that a feature at the promoter is better correlated with a DIFFERENT feature at the enhancer (example: Pol-II at the promoter vs ER-alpha at the enhancer).
- Time-lagged correlation has been used for biological network inference in the past. Here, you are not considering this. Some discussion on this would be nice (e.g. would this be computationally difficult? Would this be biologically relevant here?)
- (Somewhat related to the above) H3K4me3 levels are typically low at enhancers. Leaving out this data seems to not affect accuracy very much. Other modifications (H3K27ac?) might have been more promising. Some discussion could increase the interest of the publication.

Reviewer 2 ·

Basic reporting

The authors have developed a Bayesian classifier model which integrates genomic distance with a correlation of time-course ChIP-seq data to predict physical enhancers-promoter interactions.
The article is well written, and includes sufficient references in introduction to demonstrate the research goal. The methods are clearly described.
However, the legend of each figure should be written more carefully. For example, the definition of the shaded regions indicate in Fig 1, the error bars in Fig 3, and the number of data points in each panel in Fig 3, should be explained in legend.
Please refer MACS in the references (line 127).

Experimental design

The proposed model is evaluated using various data including ChIA-PET, Hi-C and GRO-seq data, while the scope of investigation performed is limited in enhancers-promoter interactions of stimulation-dependent genes for estrogen in MCF7 cells.
(Minor comment) Because MACS version 1.4.2 is outdated, the authors should use MACS2 for peak-calling, while the conclusion of the article may hold.

Validity of the findings

The conclusion of the article is robust in the data used for evaluation. However, the article focuses the interactions between promoters and intergenic enhancers only, while there are quite a few “genic” enhancers in the genome. Enhancer-enhancer interactions also exist. The discussion is necessary about how the proposed model works for these kinds of interaction. Furthermore, because this method can capture stimulation-dependent genes only, it should be estimated what percentage of interactions were captured by this model against all interactions captured by ChIA-PET assay.

·

Basic reporting

Complete lists of predictions should be provided as Suppl Tables.
Some texts in the figures are very small to read.

Experimental design

no comment

Validity of the findings

no comment

Additional comments

Dzida et al. developed a method to predict enhancer-promoter interactions from ChIP-seq time course data. They used ChIP-seq data of ERalpha, Pol II, H2AZ, and H3K4me3 at eight different time points after E2 stimulation in MCF7 cells, and showed, by comparing with GRO-seq for validation, that the method is able to successfully predict distally regulated target genes. In short, I find the use of time course data of several ChIP-seq experiments significantly improve the prediction of enhancer-promoter interactions in genes regulated by ERalpha. However, there are some issues the authors should address in this manuscript.

1. Pg 8, Fig 1. There is no description about blue and red lines in the figure legend. And also, why are there 12 graphs in each panel? What are the numbers at the top of each graph (I guess these are number of instances, but they are not explained in the figure legend.)? Please elaborate more on these points.
2. Pg 8, Line 313. The authors says “..., there is a substantial overlap in the positive and background distance densities due to a large separation…” (in Fig 2e). I don’t see the substantial overlap in this graph. Please clarify or explain in more detail about this. And also, what is the unit for distance (x-axis)? The numbers, especially for the exponential part, are too small to read.
3. Suppl Figs should not be numbered as 1, 2, 3, ..., but as s1, s2, s3, ....
4. Suppl Figs s1, s2, s9, s10, and S11 are not mentioned in the main text.
5. Pg 5, Line 252. I guess ‘TPR’ means ‘True Positive Rate’. Write it out in full the first time.
6. Pg 10, Fig 3a. It is hard to discriminate green and gray lines. What is the difference in thickness of these lines? X-axis must be genomic coordinate, but the unit is not clear in the figure.
7. Pg 10, Fig 3f. What is ‘3’ at the lower right corner of the ‘Pol II, ER’ panel?
8. Suppl Figs s3-6. Why are the chromosomes divided into odd and even ones?
9. Please provide complete lists of predictions as Supplementary Tables (for example, those mentioned in ‘Validation of ER-regulated target gene predictions’ in Pg 11).

---

## Round 0.2 · Minor Revisions

Your revised manuscript has been reviewed by the same three referees. As you will find, two of them now agree to accept it while the one (Reviewer 2) still requests some minor revision. Could you read the comments carefully and respond to them accordingly? I believe that your additional efforts will improve the manuscript more. Thanks for your patience, in advance.

Reviewer 1 ·

Basic reporting

The issues raised by me (and also by the other reviewers) been addressed by the authors. Especially, the clarity of the figures has been strongly improved.

Experimental design

The authors have added a detailed description of the experimental procedures of he ChIP-seq data. I think that the manuscript is acceptable for publication now.

Validity of the findings

Thanks to the improved quality of the figures and explanation of the methods, the validity of the findings can be more objectively judged. The findings of the paper are valid and suitable for publication.

Additional comments

I appreciate the efforts made in improving the quality of the manuscripts. The paper us suitable for publication now.

Reviewer 2 ·

Basic reporting

In the revised manuscript, the authors have added new experiments and explanations, and some of my concerns were satisfied, but the authors should add the discussion about the points below for publication of this manuscript. I agree with the authors that "complementary approaches to infer enhancer-promoter interactions by exploiting readily available sources of genomic data, such as ChIP-Seq and RNA-Seq data, are of interest." However, the advantages and the limitations of the proposed method against chromosome conformation capture techniques are not clearly discussed.

Experimental design

No comment

Validity of the findings

・The line 445-447 “Our model can therefore serve as a complementary approach to chromosome conformation capture techniques and offers insight into context-specific, and cell-type specific transcriptional regulation.” This method can only capture the interactions between the promoter and the intergenic enhancer for stimulation-dependent genes, while ChIA-PET and Capture Hi-C can obtain all types of interactions. Moreover, this method used ten time points for each TF, resulting in the requirement of many ChIP-seq samples, which is not cost-effective and is also technically challenging. Such limitation should be discussed in Discussion section.
・The discussion about the minimum time points required the proposed model is also necessary.
・The authors’ comment “We also limited attention to enhancers outside of gene regions. This was mainly due to the nature of our data, since we are using Pol-II as one of our features and this mark is found across the whole body of transcribing genes.” Is this a limitation of the proposed method, or the data used (Pol2)? If there is a dataset of other TF, the genic enhancers can be investigated?
・The definition of “context specific enhancer-promoter interactions” in the title and the abstract is ambiguous, and no explanation in the manuscript. If this means “stimulation-dependent enhancer-promoter interactions”, the authors should replace it.

·

Basic reporting

no comment

Experimental design

no comment

Validity of the findings

no comment

Additional comments

In the revised manuscript, the authors properly responded to all the points raised in the initial round of the review. I don't have any further concerns, and would like to recommend its publication in PeerJ.

---

## Round 0.3 · accepted · Accept

I confirm that you have appropriately re-revised the manuscript, following the comments by the reviewer.